# The Significance of Nectin Family Proteins in Various Cancerogenous Processes

**DOI:** 10.3390/ijms26073200

**Published:** 2025-03-30

**Authors:** Wiktoria Romańczyk, Anna Pryczynicz

**Affiliations:** Department of General Pathomorphology, Medical University of Białystok, 15-269 Białystok, Poland; wiktoriaustymowicz@gmail.com

**Keywords:** nectins, nectin-like molecules, cancer

## Abstract

Nectins constitute a family of Ca(2+)-independent immunoglobulin-like adhesion molecules. They are involved in cell proliferation, morphogenesis, growth, development, and immune modulation. Due to their broad involvement in physiological processes, extensive research is being conducted on the expression of individual nectins in a variety of cancers and their potential in diagnosis, prognosis, and treatment. The overexpression of nectin-1 may be a poor prognostic factor in gastrointestinal cancers (intestine and pancreas). Similarly, the overexpression of nectin-2 is a worse prognostic factor (greater tumor advancement and shorter patient survival) in cancers such as gallbladder, esophagus, and breast cancer. Changes in nectin-3 expression also affect the advancement of, e.g., colorectal cancer. Additionally, a significant factor here seems to be the change in the localization of nectin-3 expression within cellular structures. The most extensively studied nectin-4 also shows prognostic potential in many cancers. Most often, its high expression correlates with poor prognosis (e.g., gastric cancer), but it may also be a positive prognostic factor, e.g., in salivary gland cancer. Therapy based on nectin-4 is already known and used in the case of urothelial cancers. The expression of nectin-like protein 5 (necl-5) also shows prognostic and therapeutic potential in pancreatic and lung cancers, as well as in multiple myeloma.

## 1. Introduction

The nectin family consist of two groups: nectins and nectin-like molecules [1,2,3]. Through interactions with one another or with other CAMs, nectins participate in the formation of cell–cell junctions such as adherens junctions, tight junctions, and neuronal synapses. Members of the nectin family interact with other surface proteins, such as growth factor receptors, tyrosine kinase receptors, and immune receptors [4]. In this way, they are involved in many signaling pathways related to cell proliferation, morphogenesis, growth, development, and immune modulation. All of this aims to create optimal conditions that allow the organism to maintain homeostasis [5]. At the same time, their fundamental role in regulating the body’s functions means that modifications in their expression may serve as an excellent target for the development of various pathological conditions, cancers, or developmental disorders, and, in some cases, act as an entry point for viruses.

### 1.1. Nectins Family

Nectins constitute a family of Ca(2+)-independent immunoglobulin-like adhesion molecules. Based on their interactions with afadin, they are divided into two groups: nectins (nectin-1 to -4) that may interact with afadin, and nectin-like CAMs (necl-1 to -5) that do not interact with afadin [1]. The nectin members are characterized by a conserved C-terminal motif of four amino acids, which facilitates interaction with afadin. In turn, afadin binds to F-actin, thus connecting nectins to the F-actin cytoskeleton. Direct binding between nectins and afadin is mediated through the C-terminal motif of nectins and the PDZ domain of afadin. While nectins and nectins-like molecules share structural similarities, Necls do not directly bind to afadin [1,2,3]. Although necl-5 has been classified as a nectin-like molecule due to its inability to bind afadin, phylogenetically, it is more closely related to nectins than to necls. Given its properties, it could be considered to be a separate subgroup for classification purposes [2].

### 1.2. Structure

Members of the nectin family are type I single-pass membrane glycoproteins. They belong to the immunoglobulin superfamily (IgSF). They are composed of three regions: an extracellular region that includes a membrane-distal IgV-like domain and two IgC-like domains, a single transmembrane region, and a cytoplasmic region [Figure 1]. Depending on alternative splicing, each nectin may have several different isoforms that localize in specific parts of the cell [4,5].

### 1.3. Mechanism of Action

Nectin family proteins are responsible for regulating, developing, and maintaining the multicellular architecture of the organism. As membrane proteins, they ensure adhesion and cell–cell communication. They promote cross-talk at cell–cell and cell-matrix junctions [4].

#### 1.3.1. Interactions of Extracellular Regions

##### Nectins

One of the functions of nectins is the formation of cell–cell connections. The establishment of a cell–cell junction begins with the creation of a cis-dimer, consisting of two extracellular regions on the surface of the same cell. This is followed by the promotion of cell–cell contact, which may occur through the formation of either homophilic or heterophilic trans-dimers. The formation of dimers occurs independently of Ca2+. It is important to recognize that the type of binding affects its strength. Heterophilic bindings exhibit greater strength than homophilic bindings, which may determine the type of cell adhesion. All nectins are capable of forming homophilic bindings. However, the known heterophilic bindings occur exclusively between nectin-2 and nectin-3, nectin-1 and nectin-3, and nectin-1 and nectin-4 [2,4].

##### Nectine-Like Proteins (Necls)

Necls molecules have a similar structure to nectins. Their extracellular region plays a crucial role in Ca2+-independent homo- and heterophilic cell interactions. They form homophilic bindings in the form of trans-dimers, with the exception of necl-5, which forms homophilic cis-dimers. Unlike nectins, necls may form heterophilic bindings in the form of trans-dimers between all molecules within their subgroup. They also have the ability to form heterophilic connections with nectins, other ligands, and viruses [2].

#### 1.3.2. Interactions of Cytoplasmic Regions

The cytoplasmic regions of nectins interact directly with afadin, while nectins and necls interact directly with Par-3 and indirectly with annexin II, IQGAP1, and other actin-binding proteins [4].

### 1.4. Function

Nectins participate in the formation of adherence junctions (AJs) by initiating cell–cell contact through the interaction of two extracellular domains, creating a trans-binding between them. Meanwhile, the intracellular domain associates with afadin, which anchors the nectin to the actin cytoskeleton and allows for the establishment of weak cell–cell binding. Such a weak binding facilitates the recruitment of cadherins and the subsequent establishment of stable AJs [1,2,3,4].

Nectins also play a role in establishing tight junctions (TJs) by mediating intracellular signaling and initiating the reorganization of the actin cytoskeleton through binding with afadin.

Additionally, proteins from the nectin family have the ability to mediate cadherin-independent cell–cell adhesion—for example, during the process of spermatogenesis.

Nectins also initiate the formation of synaptic connections between axons and dendrites and subsequently recruit cadherins to establish a stable neuronal synapse.

In summary, nectins are responsible for the formation of organized tissue structures, and additionally, some of them, particularly necl-5, facilitate cell polarization, movement, and proliferation (1–2).

## 2. The Role of Nectins in Tumorigenesis

### 2.1. Nectin -1

Nectin-1, also known as PVRL-1, PRR-1, CD111, HveC, and HIgR, is physiologically expressed in epithelial tissues, cells of the digestive tract, liver, gallbladder, female reproductive organs, and skin. It has the ability to interact with nectin-1 (homophilic interaction), nectin-3, nectin-4, necl-1, CD96, FGFR, and integrins. The fact that it may contribute to cell adhesion formation, act as an immune receptor, and serve as a viral receptor accounts for its major functions referred to in the related literature. Dysregulation of its expression is observed in skin diseases, neurological defects, ophthalmic conditions, and mental retardation [5]. In cancerous states, the related literature reports alterations in its expression levels in breast cancer [6], pancreatic cancer [7,8], liver cancer [9], colorectal cancer [10], and cervical cancer [11]. Researchers consistently indicate that the level of nectin-1 expression increases in the cytoplasm of cancer cells [8,10,11], and an increasing number of reports provide data on the correlation between elevated nectin-1 expression and the growing metastatic potential of tumors [6,8,10]. Nevertheless, the reports are ambiguous, as some studies have also found that the loss of membrane expression at the tumor front appears to correlate with its metastatic potential [11].

Publications regarding nectin-1 expression in cancerous lesions are currently still limited. Most frequently, the literature describes studies on changes in nectin-1 expression in gastrointestinal cancers (pancreatic and colorectal cancers). It is particularly promising that those data are consistent, and researchers collectively demonstrate a significant correlation between shorter survival in patients with gastrointestinal cancers and higher nectin-1 expression [8,10]. Moreover, there are also studies on the expression of nectin in cells forming the microenvironment of gastrointestinal tumors. Yamada M. et al. examined the microenvironment of tumors in patients with diagnosed pancreatic ductal adenocarcinoma, revealing an interesting correlation between high nectin-1 expression in cancer-associated fibroblasts (CAFs) and an increased tumor metastatic potential. Additionally, their study found a high nectin-1 expression in CAFs to correlate with the presence of perineural invasion, a higher tumor stage, and even localization. Tumors in the head of the pancreas have turned out to exhibit greater nectin-1 expression compared to tumors in other locations [8]. However, data regarding the microenvironment and nectin-1 expression are inconsistent, as studies on other cancers, such as colorectal cancer, do not indicate any correlation between nectin-1 expression in the tumor microenvironment and clinicopathological parameters [10]. However, it seems that nectin-1 may have a significant potential as a marker in the diagnosis of gastrointestinal cancers. Given the consistent results regarding the correlation between its expression and survival, it seems reasonable to conduct studies on nectin-1 expression in other gastrointestinal cancers, such as gastric cancer.

Similar to gastrointestinal cancers, breast cancer also shows a trend where a high nectin-1 expression is associated with a higher frequency of metastases. However, the correlation does not reach any statistical significance. The same situation applies to the association between high nectin-1 expression and a higher stage in the TNM classification. Martin TA et al. made a remarkable accomplishment in establishing a correlation between Nectin-1 expression levels and the histopathological type of breast cancer. They demonstrated a relationship between the occurrence of ductal tumors and high nectin-1 expression. This finding opens up a discussion on the potential use of nectin-1 as a supportive marker in a differential diagnosis in histopathological diagnostics [6].

### 2.2. Nectin-2

Also known in the related literature as PVRL-2, PRR-2, CD112, and HveB, nectin-2 naturally interacts homophilically with nectin-2 and heterophilically with nectin-3, CD226, TIGIT, and CD112R. In the human body, it functions as a viral receptor for herpes simplex virus, a receptor for pseudorabies virus, and an immunological receptor, and it also mediates cell adhesion. Physiologically, it may be found in various mature tissues, including the brain, gallbladder, and pancreas, as well as in the connections between Sertoli cells and spermatids in the testes. Unlike nectin-1, nectin-2 is not observed in fibroblasts and endothelial cells [5]. The related literature indicates a correlation between the gene encoding nectin-2 and the occurrence of Alzheimer’s disease, coronary artery disease, multiple sclerosis, and cleft palate or cleft lip [12]. It is reported that alterations in its expression may be observed in neoplastic changes in breast tissue [6], ovarian tissue [13], pancreatic tissue [7], gallbladder tissue [14], colorectal tissue [15], bladder tissue [16], and squamous cell carcinomas of the esophagus [17] in acute myeloid leukemia [18,19].

In the histopathological picture of pancreatic tumors, increased cytoplasmic expression is most commonly observed, which correlates with an increase in the histological malignancy grade [8]. Similarly, in another histological type of cancer, such as squamous cell carcinoma of the esophagus, cytoplasmic expression of nectin-2 increases in poorly differentiated tumors [17]. Additionally, the literature reports the presence of membrane expression, which is initially high but gradually decreases with the reduction in the differentiation of ovarian cancer cells [13].

In the field of nectin-2 expression studies, Miao X. et al., who examined patients with squamous cell carcinoma, adenosquamous carcinoma, and gallbladder adenocarcinoma, made some interesting discoveries. They detected a significant correlation between nectin-2 expression and tumor size, an increased TNM stage, and lymph node metastasis in patients with adenocarcinoma. Furthermore, in the group of all patients, they established a correlation between high nectin-2 expression and poorer prognosis, as well as a reduced patient survival time [14]. On the other hand, the significance of nectin-2 expression for the TNM stage in esophageal squamous cell carcinoma was confirmed by Li M. et al. [17]. Additionally, a study conducted by Martin TA et al. indicated a correlation between high nectin-2 expression and a high TNM stage in breast cancer. Furthermore, the authors of the study raised the issue of more frequent metastasis in breast cancers with higher nectin-2 expression [6]. The most advanced research on nectin-2 has been conducted in patients with acute myeloid leukemia. Researchers have established not only that high levels of nectin-2 expression are associated with better prognosis in acute myeloid leukemia [19], but Stamm H. et al. also initiated preliminary studies on the potential use of immune checkpoints targeting PVRL-2 as a therapeutic option for patients [18].

On the other hand, the expression of nectin-2 in cancers such as colorectal cancer and bladder cancer does not have any clinical significance [15,16].

Such reports unequivocally confirm the diagnostic and even therapeutic potential of nectin-2. However, there are still many cancers for which there are no reports regarding the presence and impact of nectin-2 expression on their development. Therefore, this topic still requires further investigation.

### 2.3. Nectin-3

Nectin-3, also known as PVRL-3, PRR-3, or CD113, interacts with nectin-1, nectin-2, nectin-3 (homophilic interaction), PVR, necl-1, necl-2, TIGIT, PDGF.R, and integrin. Within the cell, it functions as an immunological receptor and participates in cell adhesion. It is involved in the development of the eyes, teeth, inner ear, cerebral cortex, axons, and synapses during embryonic development and also plays a role in spermatogenesis [20]. Disruptions in the expression of nectin-3 may lead to pathology in the aforementioned organs and contribute to the development of cancerous conditions in various other tissues of the body. Researchers have demonstrated deregulation of its expression in breast cancer [6], ovarian cancer [21], pancreatic adenocarcinomas [7], adenocarcinomas of the lungs [22], and colorectal cancer [15].

The authors report that increased cytoplasmic, membranous, and nuclear expression of nectin-3 plays a significant role in the pathogenesis of tumors [7,21,22]. Martin TA et al. also describe the phenomenon of shifting expression between cellular compartments (a decrease in cytoplasmic expression and an increase in nuclear expression) as the tumor progresses [6].

Xu F et al., in studies involving patients with ovarian cancer, established a significant correlation between an increased expression of nectin-3 and the rising FIGO stage and, more importantly, a lower 5-year survival rate in this patient group [21]. Contrary findings were reported by Kobecki J et al., who, in patients with colorectal cancer, described not only a significant association between a low expression of nectin-3 and tumor size but also noted that a low expression correlates with increased levels of CEA and STAGE classification [15]. Interesting conclusions were obtained by Maniwa Y et al., who, after examining a group of patients with glandular lung cancer, found that membranous expression of nectin-3 has the greatest significance for tumor development in those patients. Their study reports that greater membranous expression is associated with greater pleural invasion, advanced tumor STAGE, more frequent distant metastases, and vascular invasion into both arteries and veins. In multivariate analyses, they demonstrated that a high membranous expression of nectin-3 correlates with lower patient survival rates and more frequent disease recurrence [22].

As has been seen, the research results suggest a certain consistency, indicating that nectin-3 may influence the stage of cancer progression and have significant implications for metastasis and patient survival prognosis. These data make nectin-3 a promising diagnostic and prognostic marker. It is advisable to extend the studies on nectin-3 expression to other cancers to better understand its role and clinical potential.

### 2.4. Nectin-4

Nectin-4, also known as PVRL-4, PRR-4, IgSF receptor LNIR, and EDSS1, has been most extensively studied and described in the related literature. It has the ability to interact, inter alia, with nectin-1, nectin-4⁎, TIGIT, and the prolactin receptor. Its physiological function involves its contribution to cell–cell junctions and participation in cell adhesion. It also acts as an immunological and viral receptor [5]. Naturally, the highest expression of nectin-4 is observed in fetal tissues. However, in mature tissues, its expression is most commonly seen in the skin, esophagus, and placenta and, to a lesser extent, in the stomach, urinary bladder, and prostate. Dysregulation of nectin-4 levels is observed in squamous cell carcinoma of the head and neck [23], kidney cancer [24], breast cancer [6,25,26,27], ovarian cancer [28], bladder cancer [29], pancreatic cancer [7,30], gastric cancer [31], and cancer of other parts of the digestive tract [15,32,33].

#### 2.4.1. High Expression

In the related literature, it is most commonly reported that in cancers, there is an increase in the cytoplasmic expression of nectin-4. However, in some tumors, membrane expression of nectin-4 also appears to play a significant role. Very interesting conclusions were shared by Lattanzio R. et al., who analyzed patients with breast cancer and identified two groups based on the expression of nectin-4: the m-nectin-4 group, which shows membrane expression, and the c-nectin-4 group, which shows cytoplasmic expression. They found that higher levels of m-nectin-4 expression are significantly associated with a lower disease-free survival (DFS) rate and a lower distant recurrence-free survival (DRFS) rate in patients with luminal-A breast cancer. This finding does not apply to other histological types or the rest of the population. Multivariate analyses of DFS have demonstrated m-nectin-4 expression to be an independent prognostic factor in luminal-A tumors. Specifically, high levels of m-nectin-4 account for an independent factor affecting DRFS but do not show any significance for local recurrence-free survival (LRFS). Additionally, high levels of c-nectin-4 expression are significantly associated with a higher LRFS rate in all patients and in patients with luminal-A breast cancer. Multivariate analyses of DFS have revealed low c-nectin-4 expression to be an independent prognostic factor influencing LRFS across the entire study population. In patients with luminal-A breast cancer, the level of c-nectin-4 expression was found to be an independent prognostic factor for DFS [27]. Martin TA et al. also observed a statistically significant increase in survival in breast cancer patients with an increased expression of nectin-4 [6]. On the other hand, Rajc J. et al. studied patients with luminal B (HER2-negative) breast cancer, ductal carcinoma, and lobular carcinoma. They also observed a significant impact of nectin-4 expression on overall survival, disease-free survival, and distant relapse-free survival. Additionally, in their study, nectin-4 expression was also found to correlate with tumor size [26]. Other conclusions were reached by Fabre-Lafay S. et al., who reported a significant correlation between the histopathological type of breast tumors and nectin-4 expression. They found that a higher expression is associated with ductal tumors rather than lobular ones. Additionally, they detected a positive correlation of high nectin-4 expression with several parameters, such as EGFR, ERBB2, p53, and p-cadherin, and a negative correlation with BCL-2, estrogen receptor, GATA3, and progesterone receptor [25]. Based on the collected data, there is consistency regarding the significance of nectin-4 for patient survival. Given the data displayed by Lattanzio R. et al., in the future, it is worth considering the implementation of their research methodology. As the study indicates, more precise descriptions of expression localization may significantly impact the assessment of nectin-4 expression relevance for individual patients. Nevertheless, the collected data seem sufficient to conclude that further research on nectin-4 expression in breast cancers should focus on taking advantage of its identified diagnostic and prognostic potential. It is also worth considering initiating studies to determine therapeutic possibilities arising from nectin-4 in breast cancer patients [26].

Similar to the previously mentioned studies, in bladder cancers, researchers also make discoveries regarding the differentiation of expression in various cellular structures. Ghali F. et al. found nectin-4 to exhibit a positive membrane and cytoplasmic expression in urothelial carcinoma (UC), plasmacytoid urothelial carcinoma (UC PUC), and urothelial carcinoma with squamous differentiation (UCSD), with significantly higher cytoplasmic expression compared to membrane expression in UC PUC and UCSD types. Neuroendocrine bladder tumors (NE), on the other hand, are negative for the expression of that nectin. This study indicated not only a correlation with the localization of nectin-4 expression in various cellular structures but also its significance in differentiating histopathological types [29]. Studies on nectin-4 expression in urothelial cancer have already led to the development of a drug based on it—enfortumab vedotin. That drug is currently being tested for the treatment of advanced-stage, highly metastatic urothelial cancers [34,35]. Researchers report significant differences in the patient response to the treatment depending on nectin-4 expression in various cellular structures. Klümper N et al., in their study, indicate that patients with membrane expression of nectin-4 have poorer responses to the treatment with enfortumab vedotin and may even develop drug resistance [36]. These findings cast a new light on the previously conducted research and add significance to the previously mentioned detailed analyses of the distribution of the protein in cancer cells.

Moreover, numerous reports confirm the existence of a correlation between high expression of nectin-4 in gastrointestinal cancers (esophageal, gastric, colorectal, pancreatic) and ovarian cancer with reduced patient survival [30,31,37]. This is also the case with other clinicopathological parameters, such as a larger tumor size [7,31,37], which accounts for increased angiogenesis [30], the presence of lymph node metastases [31], a higher TNM stage [31,37], and a higher STAGE [15]. Amongst the findings that have been compiled, Mikuteit M. et al., who studied patients with renal cancer, were the only ones not to have observed any significant correlation between nectin-4 expression and survival or clinical parameters [24].

#### 2.4.2. Positive Aspects of High Expression

In some types of cancer, increased expression of nectin-4 turns out to be a positive prognostic factor. Sanders C et al. demonstrated this in their study of patients with squamous cell carcinoma (SCC) of the head and neck. They found that high expression of nectin-4 significantly contributes to increased patient survival. Additionally, they identified factors that enhance its expression, such as being non-smokers versus smokers and having higher Karnofsky performance scores (F > M and non-smoker > smoker) [23]. In the case of salivary gland cancer, Mayer M et al. found a statistically significant correlation between decreased nectin-4 expression and a higher T-stage, more frequent lymph node metastases, vascular invasion, and perineural spread [32]. Such discrepancies in the significance of nectin-4 for patient prognosis across a variety of cancer types serve as the basis for studying its expression in other types of cancers. These insights could potentially enable the application of existing therapies to new patient groups.

#### 2.4.3. Drug Interactions

In addition to the role of protein expression in tumors and its significance for patient survival and pathological–clinical parameters, it appears that nectin-4 expression may also influence tissue response to other drugs. In 2015, Das D. et al. described the relationship between an increased resistance of colorectal cancer to 5-fluorouracil treatment and an elevated expression of nectin-4 in patients [38]. These findings suggest the need for a more in-depth investigation into the role of nectin-4 in cellular processes and its correlation with the mechanisms of action of currently available therapies.

### 2.5. Necl-5

Due to its genetic and structural similarities to nectins, necl-5 is also included in this review. Necl-5 belongs to the group of nectin-like molecules due to its inability to bind afadin at its C-terminal cytoplasmic tail. However, as it has been mentioned above, it shows greater concordance with the nectin group than other nectin-like molecules. Necl-5 was discovered as a receptor for poliovirus (PVR) and has most commonly been referred to by that name in the related literature [39]. Sometimes, it is also found under the names TAGE4 and CD155. Physiologically, necl-5 interacts with nectin-3, TIGIT, CD96, CD226, vitronectin, VEGF.R, PDGF.R, and integrin. It affects cell adhesion, which modulates cell migration and proliferation. It also serves as an immunological receptor as well as the previously mentioned viral receptor [5]. Necl-5 expression is present in most tissues, although it has a weak expression in physiological tissues. However, increased expression in cancerous conditions is observed, inter alia, in colorectal cancer, lung adenocarcinoma [40], melanoma, pancreatic cancer [41], and glioma [42].

Researchers most often report an increase in both membrane and cytoplasmic expression of necl-5 [41].

In pancreatic cancer, Nishiwada S. et al. discovered that patients with tumors exhibiting high necl-5 expression have a significantly worse postoperative prognosis compared to patients with tumors showing low necl-5 expression [41]. Necl-5 is also correlated with the patient survival time, showing reduced survival rates for those with higher necl-5 expression. Similar findings were reported by Nakai R. et al. with respect to lung cancer. In the multivariate meta-analysis, they found that high necl-5 expression has been an independent negative prognostic factor for patient survival. In stage I of the disease, high necl-5 expression significantly shortened disease-free survival. Additionally, they identified a statistically significant correlation between high necl-5 expression and a higher TNM stage, as well as more frequent metastasis to lymph nodes [40]. Researchers also demonstrated that necl-5 expression is correlated with VEGF expression in tumors and with tumor vascular density [41]. Within the framework of their study, Lee BH et al., having studied multiple myeloma, reported a correlation between high necl-5 expression and an increased disease stage in the Revised International Staging System (R-ISS) [43]. Based on the research, necl-5 was found to have significant diagnostic potential, which could be further explored in other types of cancer.

#### Clinical Use of Necl-5

Research is also emerging on the use of necl-5 in the treatment of cancerous lesions, with promising results reported, inter alia, by Li S. et al., who observed a significant positive impact arising from blocking necl-5. They specifically noted a reduction in the invasion of medulloblastoma cells and an increase in their mortality [44]. Necl-5 and nectin-2 are being investigated as potential therapeutic targets in acute myeloid leukemia (AML) [18]. Necl-5 thus exhibits not only diagnostic potential but also proves to be a promising target for cancer therapies. It is important to continue expanding the related research and assess its potential in other types of cancers.

The most important findings on the role of nectins in various cancers are summarized in Table 1.

## 3. Conclusions

A comprehensive review of the extant scientific literature reveals the multidirectional potential of nectins as possible diagnostic and prognostic markers, as well as a basis for the development of anticancer therapies. A growing body of evidence suggests that nectin-1 demonstrates prognostic and diagnostic potential in breast, colorectal, and pancreatic cancers. Further studies are required to investigate the expression of nectin-1 in other cancers and to ascertain whether it exhibits a similar capacity for diagnosis and prognosis. Currently, there is a paucity of data on nectin-1 as a target for anticancer treatment, and its potential in this regard remains to be elucidated. In contrast, nectin-2 is currently being investigated for its potential use in the treatment of acute myeloid leukemia, and the results appear to be promising. Furthermore, nectin-2 also has the potential to be a prognostic and diagnostic marker in cancers such as gallbladder cancer, pancreatic cancer, oesophageal cancer, breast cancer, and acute myeloid leukemia. However, further research is required to identify changes in its expression in other cancers and their correlation with prognostic and diagnostic factors. The identification of changes in nectin-3 expression in various cancers will provide a foundation for the development of novel anti-cancer therapies that target cellular pathways using nectin-3. Scientific reports on nectin-3 are also quite limited; however, the available data clearly indicate that nectin-3 influences tumor progression in colorectal cancer, ovarian cancer, and lung cancer, as well as has a significant prognostic impact on patient survival. These data make it a promising diagnostic and prognostic marker. It is recommended that the study of nectin-3 expression be expanded to encompass additional cancers. Nectin-4, given the number of reports and the diversification of topics, has attracted the most attention from researchers to date. Nectin-4 has been identified as having diagnostic, prognostic, and therapeutic potential, and therapy of urothelial cancers based on the mechanism of action of nectin-4 is already known and in use today. In the context of breast cancers, the expression of nectin-4 appears to be sufficiently well understood that, in addition to the potential utilization of the data for prognostic and diagnostic purposes, there is a strong likelihood of the emergence of attempts at therapeutic use in the near future. The potential for therapeutic exploitation of nectin-4 is particularly promising, with the expression of this protein correlating with patient prognosis in various tumor types. This underscores the necessity for a comprehensive investigation of nectin-4 expression across a wide range of cancer types. In addition to urothelial carcinoma of the bladder, there is significant potential for vedotin enfortumab to be used in other cancers. Nectin-5 demonstrates a robust prognostic potential in pancreatic and lung neoplasms. Recent studies on the use of nectins in medulloblastoma and acute myeloid leukemia therapies have generated a positive outlook for future research in this field. The data presented in the aforementioned review indicates that further investigation into the therapeutic potential of nectins is warranted, particularly in relation to other cancers that have not yet been explored. A particular emphasis should be placed on determining the specific location of intracellular expression, which seems to be the key information to determining the involvement of nectins in tumor transformation. This may not only contribute to better systematization but also to the establishment of diagnostic–prognostic–therapeutic models, which, with this knowledge, can be aptly applied to a larger group of patients.

## Figures and Tables

**Figure 1 ijms-26-03200-f001:**
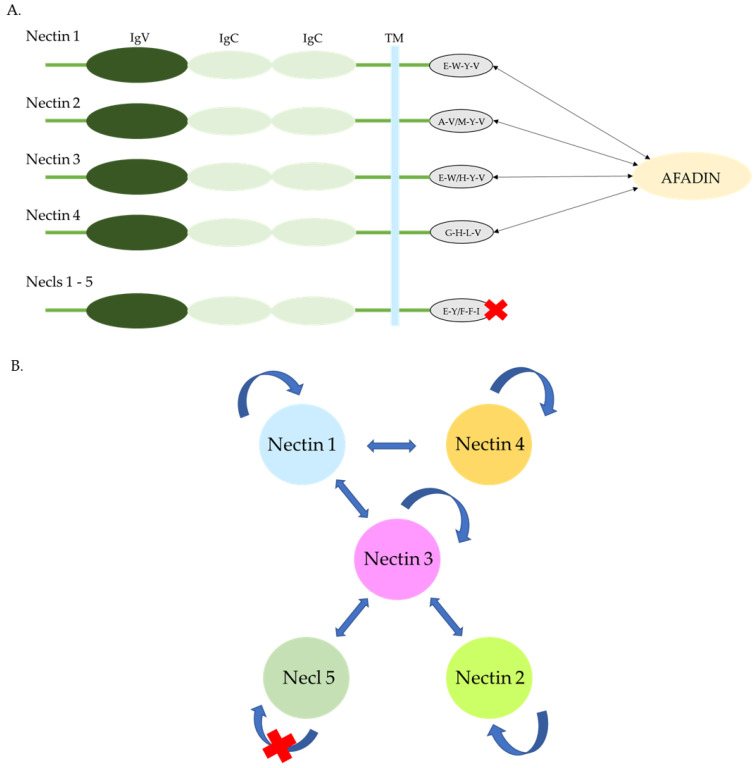
**(A**) Structure of nectins and necls. Nectins and necls have a similar structure, with extracellular domains consisting of one IgV and two IgC regions, a transmembrane region (TM), and a cytoplasmic tail containing an afadin-binding motif in nectins. (**B**) Interactions of nectins and necl-5. Rounded arrows indicate homophilic interactions. Straight arrows indicate heterophilic interactions.

**Table 1 ijms-26-03200-t001:** Nectin expression and cancers.

	Cancer Type	Most Important Correlation of Protein Overexpression with Clinicopathological Parameters	References
Nectin-1			
	Colorectal cancer	shorter survival	[10]
	Pancreatic cancer	shorter survival	[8]
	Breast cancer	histopathological type	[6]
Nectin-2			
	Pancreatic cancer	histological malignancy grade	[8]
	Gallbladder adenocarcinoma	tumor size higher TNM stage lymph node metastasis poorer prognosis shorter survival	[14]
	Gallbladder squamous cell carcinoma and adenosquamous carcinoma	poorer prognosis shorter survival	[14]
	Brest cancer	higher TNM stage	[6]
	Acute myeloid leukemia	better prognosis	[19]
Nectin-3			
	Ovarian cancer	higher FIGO stage shorter survival	[21]
	Colorectal cancer	lower levels of CEA lower STAGE	[15]
	Glandular lung cancer	higher pleural invasion higher STAGE more frequent distant metastases shorter survival more frequent disease recurrence	[22]
Nectin-4			
	Breast cancer (luminal-A)	shorter disease-free survival shorter distant recurrence-free survival	[27]
	Breast cancer	longer survival histopathological type	[6] [25]
	Bladder cancers	histopathological type	[29]
	Gastric cancers	shorter survival larger tumor size more frequent lymph node metastases higher TNM stage	[31]
	Pancreatic cancers	shorter survival larger tumor size increased angiogenesis	[30]
	Squamous cell carcinoma of head and neck	longer survival	[23]
	Salivary gland cancer	lower TNM stage less frequent lymph node metastases less frequent vascular invasion less frequent perineural spread	[32]
Necl-5			
	Pancreatic cancer	shorter survival	[41]
	Lung cancer	shorter survival shorter disease-free survival higher TNM stage more frequent metastasis to lymph nodes	[40]

## Data Availability

Not applicable.

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
