# Peer review of "The Significance of Nectin Family Proteins in Various Cancerogenous Processes"

_ijms, 2025, doi:10.3390/ijms26073200_

Round 1
Reviewer 1 Report
Comments and Suggestions for Authors
This short manuscript presents an intriguing approach to reviewing nectins as a family of adhesion receptors with a still poorly characterized function in cancer. Despite the high number of reviews on similar topics, most focusing on nectin-4 as a key molecule in the pathogenesis and prognosis of breast and genitourinary cancer as well as other solid tumours, this manuscript remains interesting. Overall, the manuscript is well written, but several important points should be addressed before it can be considered for publication in IJMS:
- Typographic Errors: Carefully proofread the title and other sections of the manuscript, including the repeated paragraph at the beginning of the “Introduction” section and the “Nectins family” subheading.
- Figure 1: The figure seems too simplistic. Provide a detailed representation of the structural and functional motifs of nestins and their intracellular and extracellular ligands.
- Afadin's Role: The role of afadin in nestins signaling and function should be better explained.
- Nectin-Related Discussion: The manuscript coments on reports showing the relationship between altered expression of nectins in cancer patients and tumorigenesis, offering insights into their potential as biomarkers and in cancer prognosis. However, there is limited discussion on how nectins could be involved in these processes. For instance, what could be the mechanism of action of nectin-4 in resistance to 5-fluorouracil treatment in cancer patients? An extended discussion on this and other roles of nectins in tumorigenesis is needed.
- Conclusion section: This section should be rewritten. The key points of the review, i.e., the correlation between the expression levels of some nectin members and cancer, should not be merely listed, but properly integrated and discussed in the corresponding paragraph. In addition, new perspectives for the use and function of nectins in cancer may be proposed.
Author Response
Comments 1: Typographic Errors: Carefully proofread the title and other sections of the manuscript, including the repeated paragraph at the beginning of the “Introduction” section and the “Nectins family” subheading.
Response 1: Thank you very much for your comment, corrections have been done in the introduction of the original text.
Comments 2: Figure 1: The figure seems too simplistic. Provide a detailed representation of the structural and functional motifs of nestins and their intracellular and extracellular ligands.
Response 2: Thank you very much for your comment, the new figure has been placed.
Comments 3: Afadin's Role: The role of afadin in nestins signaling and function should be better explained.
Response 3: Thank you very much for your comment, we have included the following information in the text.
"]. The Nectins members are characterised by a conserved C-terminal motif of four amino acids, which facilitates interaction with afadin. In turn, afadin binds to F-actin, thus connecting nectins to the F-actin cytoskeleton. Direct binding between nectins and afadin is mediated through the C-terminal motif of nectins and the PDZ domain of afadin. While nectins and nectins-like molecules share structural similarities, Necls do not directly bind to afadin [1-3]. "
Comments 4: Nectin-Related Discussion: The manuscript coments on reports showing the relationship between altered expression of nectins in cancer patients and tumorigenesis, offering insights into their potential as biomarkers and in cancer prognosis. However, there is limited discussion on how nectins could be involved in these processes. For instance, what could be the mechanism of action of nectin-4 in resistance to 5-fluorouracil treatment in cancer patients? An extended discussion on this and other roles of nectins in tumorigenesis is needed.
Thank you very much for this comment. Our work focused on the differences in nectin expression in individual cancers, less on their impact on particular treatments. The information regarding the existence of an effect of nectin-4 expression on patient treatment was included to show a potential direction for research in the case of nectin-4 and other nectins. Based on the available data in the pub-med database on which our review was based, under the searches ‘nectin-4’, ‘5-fu’, 5-fluorouracile we see only two scientific papers in which the researchers rely on material taken from patients' tumour lesions. In the face of such poor data, it is difficult to develop a discussion on this topic, especially as the authors of the study Trop-2 and Nectin-4 immunohistochemical expression in metastatic colorectal cancer: searching for the right population for drugs' development, which we did not cite in the text of our paper, themselves cast doubt on their results due to the lack of repeated assessments of nectin-4 expression on different slides of the same tumour.
Comments 5: Conclusion section: This section should be rewritten. The key points of the review, i.e., the correlation between the expression levels of some nectin members and cancer, should not be merely listed, but properly integrated and discussed in the corresponding paragraph. In addition, new perspectives for the use and function of nectins in cancer may be proposed.
Response 5: Thank you very much for your comment, the conclusions section has been re-edited.

Reviewer 2 Report
Comments and Suggestions for Authors Romańczyk et al.'s paper "The significance of nectin-family proteins in various cancerogenous processes" is intriguing. The nectins are important in cancer development. To increase its value, more integration and literature review will be necessary.
1. The “Mechanism of Action” need be expanded well. Normal functions for nectins; the possible common mechanism in pathological development etc. A summary scheme will be nice for a review.
- As you mentioned in your manuscript, nectin overexpression may lead to positive or negative aspects in cancers, were those because of various signal transduction pathways in development in various organs or cancers. Are there any common or unique pathways for various organs or cancers? Do you have any recommendations for further study or observations?
- Future research on the application and mechanism of nectins etc could be discussed more extensively.
Author Response
Comments 1: The “Mechanism of Action” need be expanded well. Normal functions for nectins; the possible common mechanism in pathological development etc. A summary scheme will be nice for a review.
Response 1: The paragraph on interaction mechanisms is intended merely to provide the reader with a general overview of the localisation of nectins and their mechanisms of action. However, as the focus of our work has been on presenting the clinical aspects of nectin expression, it has not been expanded to include the precise mechanisms of action, in order to avoid overwhelming the reader with information that will not be considered subsequently.
Comments 2: As you mentioned in your manuscript, nectin overexpression may lead to positive or negative aspects in cancers, were those because of various signal transduction pathways in development in various organs or cancers. Are there any common or unique pathways for various organs or cancers? Do you have any recommendations for further study or observations?
Response 2:In conducting the review for the following publication, we noted that there are reports of differential pathways, but as we were interested in aspects of immunohistochemical expression in our work, cellular mechanisms were not the focus of our study. Whereas we have included suggestions for expanding research on nectins in a new summary.
Comments 3: Future research on the application and mechanism of nectins etc could be discussed more extensively.
Response 3:Thank you for your comment, we have included a new conclusion section.

Reviewer 3 Report
Comments and Suggestions for Authors
The authors present the manuscript entitled "The significance of nectin-family proteins in various cancerogenic processes" in which they present a review on the participation of the nectin family in various cellular processes associated with different types of cancer.
I consider the review to be interesting and quite clear, however, I have some comments:
1. Lines 26 to 47 are duplicated. The information must be reviewed and the duplicates removed.
2. I consider that a figure on the interactions and functions of nectins would help to better understand this part of the manuscript.
3. If a figure could be included with the possible mechanisms or cellular processes associated with cancer in which the involvement of nectins with the different types of cancer or a general mechanism that explains their participation is detailed, it would help to better understand the review.
4. In lines 342-343 "Necl-5 expression is present in most tissues, although it is poorly detectable in physiological tissues. ", I don't understand this sentence, are they referring to physiological conditions?
5. Are there studies where the expression of different nectins is analyzed at the same time in the same tissues or the same samples of cancerous tissues to look for a correlation of different expression profiles of these proteins and associate them with prognosis or patient survival.
6. For all the information included in this review, perhaps the number of references cited is small.
Comments on the Quality of English LanguageThe manuscript must be revised to correct minor errors in English writing.
Author Response
Comments 1. Lines 26 to 47 are duplicated. The information must be reviewed and the duplicates removed.
Response 1: Thank you for your comment we have re-edited the mentioned part in the introduction.
Comments 2. I consider that a figure on the interactions and functions of nectins would help to better understand this part of the manuscript.
Response 2: Thank you for your comment, we established a new figures.
Comments 3. If a figure could be included with the possible mechanisms or cellular processes associated with cancer in which the involvement of nectins with the different types of cancer or a general mechanism that explains their participation is detailed, it would help to better understand the review.
Response 3:
Thank you for your comment, Our work focused on clinical aspects and histochemical expression in patient material. Conducting such detailed analyses of the mechanisms of nectin action in healthy and tumour-affected tissues is such a major subject, requiring very thorough research, which we believe could be the subject of another publication.
Comments 4. In lines 342-343 "Necl-5 expression is present in most tissues, although it is poorly detectable in physiological tissues. ", I don't understand this sentence, are they referring to physiological conditions?
Response 4: Thank you for your comment, we have rewroted the sentens
„Necl-5 expression is present in most tissues, although it has a weak intensity pattern in physiological tissues.” propably it was a mistake of the translation.
5. Are there studies where the expression of different nectins is analyzed at the same time in the same tissues or the same samples of cancerous tissues to look for a correlation of different expression profiles of these proteins and associate them with prognosis or patient survival.
Thank you for your comment, we recomend to look closely to :
Martin, T.A.; Lane, J.; Harrison, G.M.; Jiang, W.G. The expression of the Nectin complex in human breast cancer and the role of Nectin-3 in the control of tight junctions during metastasis. PLoS One 2013, 8, e82696. doi: 10.1371/journal.pone.0082696,
Izumi, H.; Hirabayashi, K.; Nakamura, N.; Nakagohri, T. Nectin expression in pancreatic adenocarcinoma: nectin-3 is associated with a poor prognosis. Surg. Today 2015, 45, 487–494. doi: 10.1007/s00595-015-1126-2.
6. For all the information included in this review, perhaps the number of references cited is small.
Thank you for your comment, for our review, we used the pubmed search algorithm, which involved performing searches under the keywords “nectin-1” “nectin-2”, ‘nectin-3’, ‘nectin-4’ and ‘ncls-5’ in correlation with the keywords ‘cancer’ and ‘immunohistochemistry’, the works cited in our review constitute the entire database of publications available in the pubmed browser. From the total number of cited publications we discarded those in which research was carried out on animals or cell cultures, as our point of interest is the immunohistochemical study of scrapings from human tumors. We are sorry that the number of citations in your opinion is insufficient, but this is the total number of records that can be obtained on this topic from the pubmed database.

Round 2
Reviewer 1 Report
Comments and Suggestions for Authors
The authors have addressed all my major comments, but proofreading is still required to improve the quality of the paper.
Some examples are the following:
-"Significance" instead of "Sagnificance" in the title.
- Line 384: The word work/study/review seems more appropriated than "thesis".
- Rewrite/correct the sentence from line 410 to 412. The way it has been written seems misleading.
Author Response
Thank you for your comments, we are enclosing new version.

Reviewer 2 Report
Comments and Suggestions for Authors
I am fine with the responses.
Author Response
Thank you